# One-year experience with latanoprostene bunod ophthalmic solution 0.024% in clinical practice: A retrospective observational study

**Chun-Mei Hsueh**[1,2], **Chen-Hsin Tsai**[1,2], **Jou-Chen Huang**[1,2], **Si-Huei Lee**[1,2], **Tsung-Jen Wang**[1,2], **Siao-Pei Guo**[1,2]*

1 Department of Ophthalmology, Taipei Medical University Hospital, Taipei Medical University, Taipei, Taiwan, 2 Department of Ophthalmology, School of Medicine, College of Medicine, Taipei Medical University, Taipei, Taiwan

* d825107001@tmu.edu.tw

**Data Availability Statement:** All relevant data are within the manuscript and its Supporting Information files.

## Abstract

### Purpose

We evaluated the IOP-lowering efficacy and safety of latanoprostene bunod (LBN) ophthalmic solution 0.024% (Vyzulta®), the first topical nitric oxide-donating prostaglandin analog (PGA), in clinical practice.

### Materials and methods

A retrospective medical chart review from July 2021 to July 2023 of patients with open-angle glaucoma receiving LBN with at least 1 year follow–up was conducted. All included patients received LBN 0.024% as a replacement for a PGA, with examinations at 1-, 3-, 6-and 12-months follow-up. Main outcome measures were IOP, retinal nerve fiber layer thickness, visual fields before/after LBN use and adverse effects. Subgroup analysis with glaucoma types and PGA use were performed for additional IOP reduction after LBN use.

### Results

Among 78 included patients, 47 patients (81 eyes), 60% with open-angle glaucoma (OAG) remained on LBN throughout 12-month follow-up. Baseline IOP was 18.2±4.2 mm Hg, and Prostaglandin analog (PGA)-IOP was 14.4 ± 3.0 mm Hg (21% mean IOP reduction). After switched to LBN, mean additional IOP reduction was 1.0 mm Hg at month 1, and the greatest reduction was 1.6 mm Hg (8.8% additional mean IOP reduction) at month 12 (P<0.0001). Subgroup analysis (NTG, 73%) showed that mean additional IOP reduction at month 12 was 1.3±2.0 mm Hg in NTG group and 2.1±3.2 mm Hg in POAG group (7.7% vs. 8.7% additional IOP reduction rates, P = 0.23). Subgroup analysis of PGA use at month 12 was 1.8±2.3 mm Hg in tafluoprost group and 0.5±1.7 mm Hg in travoprost group (9.5% vs.2.6% additional IOP reduction rates, P = 0.02). Tolerable ocular adverse effects included irritation (n = 16, 19.8%), mild conjunctival hyperemia (n = 11, 13.6%), dark circles (n = 4, 4.9%) and blurred vision (n = 2, 2.5%). There were no significant visual field and retinal nerve fiber layer thickness changes after 12 months of treatment with LBN 0.024%.

**Funding:** The author(s) received no specific funding for this work.

**Competing interests:** The authors have declared that no competing interests exist.

## Conclusions

Although high intolerable adverse effects including conjunctival hyperemia and eye irritation happened in the first month, remaining sixty percent of patients exhibited statistically significant additional IOP reductions in the replacement of other PGAs during 12 months of clinical use of LBN 0.024%.

## Introduction

Glaucoma is the leading cause of irreversible blindness. The prevalence of primary open-angle glaucoma (POAG) in Asian populations is between 1.0% and 3.9%, and normal tension glaucoma (NTG) comprises the majority (46.9%-92%) of open-angle glaucoma (OAG) in Asian epidemiologic studies, whereas the calculated mean proportion of NTG is 33.7% in white populations [1]. IOP is an important risk factor for glaucoma progression. The Early Manifest Glaucoma Trial showed that every 1mmHg reduction in IOP yielded a 10% reduction in the risk of visual field progression [2]. Two main classes of topical ocular hypotensive agents are used commonly for the treatment of OAG or ocular hypertension (OHT). The first class includes beta-adrenergic receptor antagonists, carbonic anhydrase inhibitors and alpha-adrenergic receptor agonists, which lower IOP by reducing aqueous humor production. The other class of ocular hypotensive agents is prostaglandin analogs, including latanoprost, which increase aqueous humor outflow facilities through uveoscleral routes and possibly by trabecular meshwork. To date, pharmacologic reduction of IOP has remained the standard initial treatment of OAG or OHT [3].

Latanoprostene bunod (LBN) 0.024% (trade name, Vyzulta®) is a new IOP-lowering eye drop approved by the US Food and Drug Administration (FDA) in November 2017. LBN is a dual-mechanism, dual-pathway molecule, consisting of latanoprost acid, which is known to enhance uveoscleral outflow (unconventional outflow pathway) by upregulating matrix metalloproteinase expression and remodeling the extracellular matrix of the ciliary muscle; it is linked to an NO-donating moiety, which enhances trabecular meshwork/Schlemm's canal (conventional) outflow by inducing cytoskeletal relaxation via the soluble guanylyl cyclase-cyclic guanosine monophosphate (sGC-cGMP) signaling pathway. Both latanoprost and NO are known to reduce IOP in humans [4]. In phase 3 randomized controlled trials (RCTs), APOLLO [5] and LUNAR [6],LBN delivered a 32% mean IOP reduction and up to 9 mmHg reduction from baseline in patients with open-angle glaucoma (OAG) and ocular hypertension (OHT); it also provided greater IOP-lowering ability compared with timolol 0.5% twice-a-day and maintained the reduced IOP through 12 months [7]. In the Jupiter study [8], LBN reduced the mean IOP 5.3 mmHg (25%) in patients with normal tension glaucoma (NTG) after 1 year treatment.

In a meta-analysis of the short-term efficacy of LBN for treatment OAG and OHT, LBN 0.024% solution revealed greater IOP-lowering effect than latanoprost and travoprost and was similar to the effects of bimatoprost 0.01%. LBN demonstrated a safety profile comparable to that of conventional PGAs [9]. The real-world impact of LBN 0.024% in the replacement of other PGAs and effects in the visual field and retinal nerve fiber layer thickness preservation have rarely been investigated. So we conducted a retrospective chart review study to evaluate the IOP-lowering efficacy and safety of LBN 0.024% in the replacement of other PGAs with at least 12 months follow-up.

## Materials and methods

Data were retrospectively collected from electronic medical records from July 2021 to July 2023 of Taipei Medical University Hospital since September 2023. Inclusion criteria were patients aged 18 years or older with OAG (including NTG) who received PGA treatment, had best-corrected visual acuity of 20/40 or better, eyes had open angles on gonioscopic examination. OAG was defined as an eye with glaucomatous optic nerve head change and corresponding glaucomatous visual field defects. If both eyes of the same patient were found to be eligible, both eyes were selected for analysis. Patients with diseases known to affect the visual field (eg, pituitary lesions, Alzheimer disease, stroke, diabetic retinopathy) or inability to perform perimetry reliably, or with life-threatening chronic diseases, were excluded. All included patients received LBN 0.024% as a replacement for a PGA, with examinations at 1-, 3-, 6-and 12-months follow-up. After beginning treatment, no surgeries, lasers, or medication changes occurred during follow-up. Main outcome measures were IOP, retinal nerve fiber layer thickness, visual fields before and after LBN use, and adverse effects.

All participants received complete ophthalmic examinations, including best-corrected visual acuity, gonioscopy, pachymetry using specular microscope (EM 3000, Tomey, USA), sli-tlamp biomicroscopy and stereoscopic disc photography (Canon. CR-2 AF). Intraocular pressure measurements used non-contact tonometer (TONOREF III, NIDEK, Japan) and rechecked with iCare tonometer (TA01i) for 3 times. If IOP difference greater than 3 in three times measurements with non-contact tonometer, then recheck with iCare tonometer for three times. If the average values of two kinds of measurements are different, the average values of IOP data from iCare tonometer was recorded. The method of IOP measurement used for each patients was consistent across visits. Spectral-domain OCT (Heidelberg Spectralis OCT, Heidelberg Engineering, Germany); and a visual field (VF) test (Kowa AP-5000c; KOWA, Los Angeles, CA, USA). OCT and VF examinations before and after the LBN use will be recorded. Only reliable VF test results (false-positive errors <15%, false-negative errors <15%, and fixation loss <20%) were included in the analysis. Peripapillary RNFL thickness was measured using Spectralis SD-OCT (Heidelberg Engineering, Germany). SD-OCT scans acquire a total of 1536 A-scan points from a 3.45-mm circle centered on the optic disc. Images with non-centered scans or signal strength 15 or less were excluded.

## Ethics statement

The study design followed the principles of the Declaration of Helsinki and the study protocol was approved by the Ethics Committee of the Taipei Medical University (TMU-N202210003), which also acts on behalf of its affiliated hospital, Taipei Medical University Hospital. Since all data were retrospective and included patients were deidentified, the Ethics Committee waived signed informed consent.

## Statistical analysis

Patient and eye characteristics were determined using the independent t test for continuous variables and the Chi squared test for categorical variables. The differences of IOP reduction between baseline IOP, PGA IOP and IOP with LBN 0.024% treatment at 1-, 3-, 6- and 12-months were calculated using linear mixed-effects models, with fixed effect terms for treatment and period, and a within-participant random intercept. Visual field mean deviation (MD), pattern standard deviation (PSD) and retinal nerve fiber layer (RNFL) thickness before and after treatment with LBN for 12 months were compared with the paired t test. In subgroup analysis of patients who received tafluprost or travoprost at baseline, and POAG or NTG, the independent t test was performed. Differences were determined using two-sided 95%

Confidence Intervals (CI) and P values were calculated. P < .05 was considered statistically significant. All statistical analyses were performed using SPSS (version 26.0; SPSS Inc, Chicago, IL, USA)

## Results

Among seventy-eight included patients, 47 patients (81 eyes, 60%) with open-angle glaucoma (OAG) remained on LBN treatment throughout 12-months follow-up. 31 patients (40%) dropped out for intolerable adverse effects after 1 month use of LBN, and the most common ocular side effects were irritation and conjunctival hyperemia. Patients' eyes with POAG (n = 22, 27.2%) and NTG (n = 59, 72.8%) were included. Patients' demographic and clinical characteristics are shown in Table 1. Baseline IOP was 18.2 ± 4.2 mm Hg and Prostaglandin analogue (PGA)-IOP was 14.4 ± 3.0 mm Hg, on 1.3 ± 0.5 glaucoma medications. PGA use before LBN included tafluoprost 0.0015% (n = 46, 56.8%), travoprost 0.003% (n = 31, 38.3%) and latanoprost 0.005% (n = 4, 4.9%). 63% patients were receiving PGA monotherapy. The mean additional IOP reduction (95% CI) after LBN use at 1-, 3-, 6- and 12-month follow-up were 1.0 ±0.2 mmHg, 1.3±0.2 mmHg, 1.1±0.2 mmHg and 1.6 ±0.2 mmHg (P<0.001). Differences in IOP reduction were all statistically significant, with the greatest reduction at 12-month follow-up. (Table 2)

Subgroup analysis of patients who had POAG or NTG at baseline showed that mean additional IOP reduction was significantly different after 1 and 3 months (S1 Table) In the subgroup who was POAG (baseline IOP: 24.7±2.9 mm Hg; n = 22; 27%), mean PGA-IOP was 16.8±3.5 mm Hg (32% mean IOP reduction). In the subgroup who was NTG (baseline IOP: 16.8±2.3 mm Hg; n = 59; 73%), mean PGA-IOP was 13.5±2.1 mm Hg (19.6% mean IOP

**Table 1. Characteristics of participants (n = 81 eyes).**

| | |
|---|---|
| SE (Diopter) | -6.0 ± 3.3 (-5.3 - -6.8) |
| Age (y) (95% CI) | 51.3 ± 11.9 (48.6–54.0) |
| Sex, Female (%) | 30 (37.04) |
| CCT (um) | 541.3 ± 34.7 (533.0–549.6) |
| Diabetes, n (%) | 9 (11.4) |
| Hypertension, n (%) | 24 (29.6) |
| Hyperlipidemia, n (%) | 28 (35.4) |
| Baseline IOP (mmHg) (95% CI) | 18.2 ± 4.2 (17.2–19.2) |
| PGA- IOP (mmHg) | 14.4± 3.0 (13.7–15.0) |
| Pre-LBN RNFL (um) | 71.9± 10.5 (69.4–74.4) |
| Pre-LBN VF MD (dB) | -3.4±1.6 (-10.0- -0.2) |
| Pre-LBN VF PSD (dB) | 3.7±2.8 (0–10.2) |
| Eye classification, n (%) | |
| POAG | 22 (27.2) |
| NTG | 59 (72.8) |
| PGA use before LBN (%) | |
| tafluprost | 46 (56.8) |
| travoprost | 31 (38.3) |
| latanoprost | 4 (4.9) |

CCT = central corneal thickness; CI = confidence interval; IOP = intraocular pressure; PGA = prostaglandin analogue; RNFL = retinal nerve fiber layer thickness; POAG = primary open angle glaucoma; NTG = normal tension glaucoma; LBN = latanoprostene bunod; Mo = month

**Table 2. Mean additional intraocular pressure (IOP) reduction after LBN use (n = 81eyes).**

| | | |
|---|---|---|
| PGA-IOP vs IOP 1Mo | -1.0±0.2 (0.4–1.6) | <0.0001 |
| PGA-IOP vs IOP 3Mo | -1.3±0.2 (0.7–1.9) | <0.0001 |
| PGA-IOP vs IOP 6Mo | -1.1±0.2 (0.5–1.7) | <0.0001 |
| PGA-IOP vs IOP 12Mo | -1.6±0.2 (1.0–2.2) | <0.0001 |

IOP = intraocular pressure; LBN = latanoprostene bunod; PGA = Prostaglandin analogue; Mo = month

$P<0.05$ is considered statistically significant

reduction). The mean additional IOP reduction from PGA-IOP was 2.1±3.2 mm Hg (8.7%) and 1.3±2.0 mm Hg (7.7%) after switched to LBN in POAG and NTG group at month 12. (Fig 1)

Subgroup analysis of patients who used tafluprost or travoprost before LBN showed that additional mean IOP reduction was significantly different at 1- and 12-month follow-up visits. (S2 Table) Mean IOP reduction were 21% and 22% in tafluprost and travoprost group from baseline. Mean additional IOP reduction at month 12 after switched to LBN was 1.8±2.3 mm Hg (9.5%) in tafluprost group and 0.5±1.7 mm Hg (2.6%) in travoprost group (P = 0.02). (Fig 2)

The differences between visual field mean deviation (-0.04 ± 1.2, P = 0.82) and pattern standard deviation (-0.2 ± 1.9, P = 0.42) after switched to LBN for 12 months were not statistically significant. The differences in RNFL thickness (0.1 ± 4.4, P = 0.93) was also not statistically significant after 12 months of treatment with LBN 0.024%.

Approximately 40% of patients have intolerable ocular adverse effects, mostly moderate to severe conjunctival hyperemia and instillation site pain, so they dropped out at 1 month of

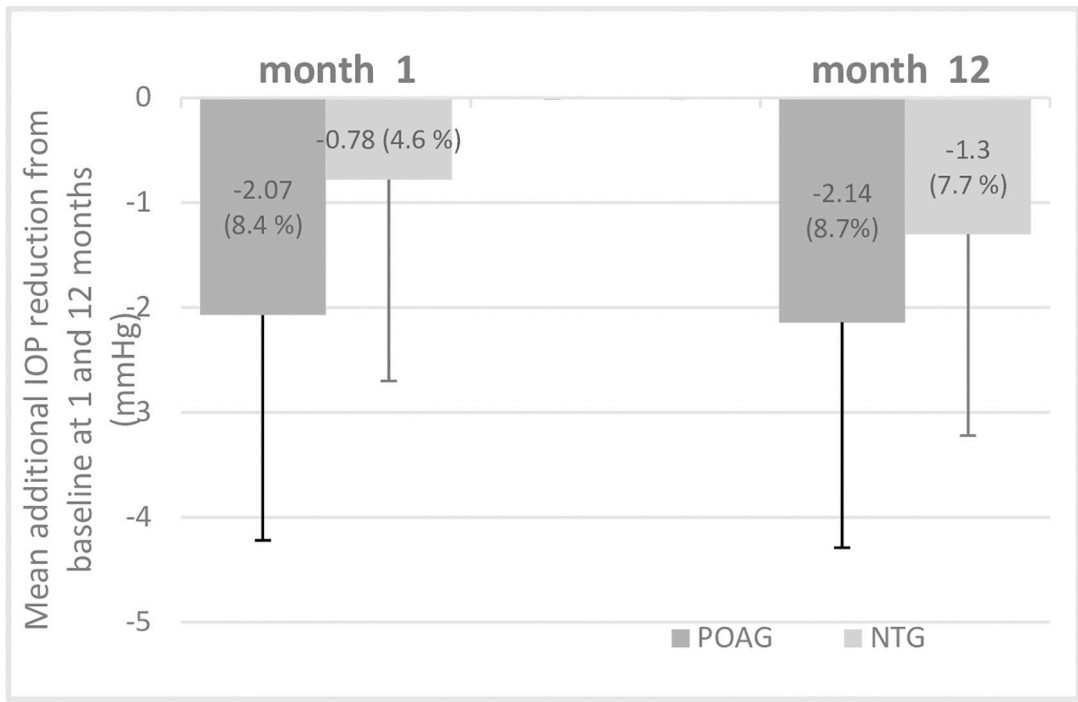

**Fig 1. Mean additional intraocular pressure (IOP) reduction at month 1 and month 12 in patient subgroups who were primary open angle glaucoma (POAG, n = 22) or normal tension glaucoma (NTG, n = 59).** P = 0.03 at month 1 and P = 0.23 at month 12.

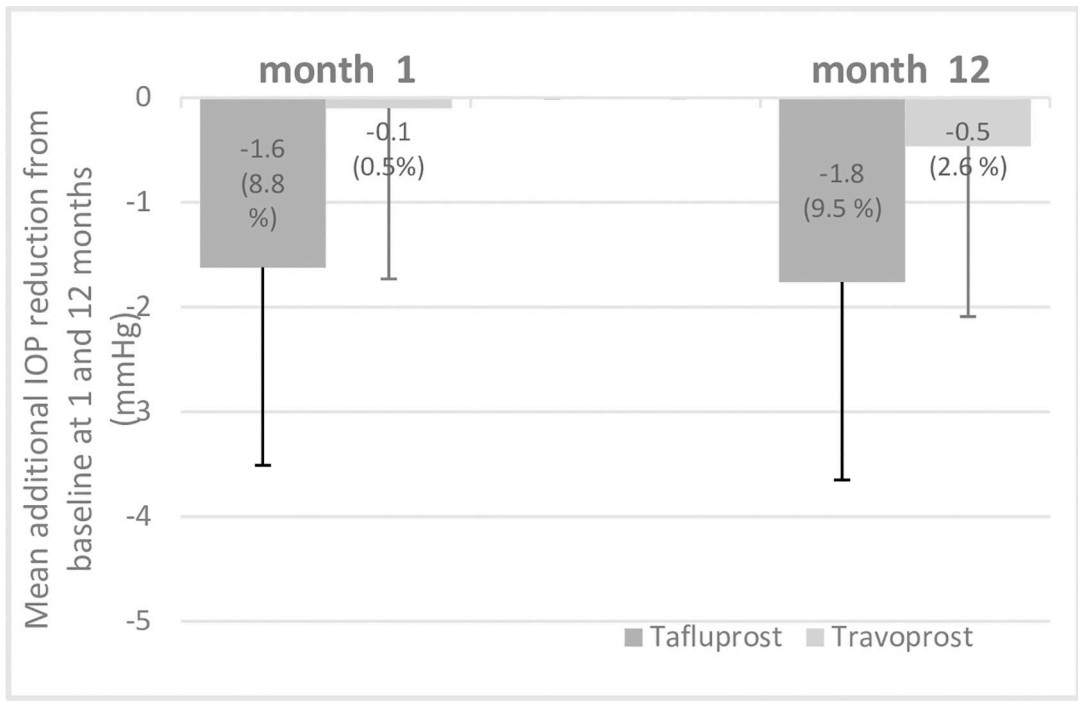

**Fig 2. Mean additional intraocular pressure (IOP) reduction at month 1 and month 12 in patient subgroups previously treated with tafluprost (n = 46) or travoplost (n = 31).** P = 0.001 at month 1 and P = 0.02 at month 12.

LBN use and changed medication. Tolerable ocular adverse effects were 40.8% in remaining 81 eyes included irritation (n = 16, 19.8%), mild conjunctival hyperemia (n = 11, 13.6%), dark circles (n = 4, 4.9%) and blurred vision (n = 2, 2.5%).

## Discussion

In this real-world retrospective chart review study, there was 21% mean IOP reduction from baseline treatment with PGA. After switched to LBN, there was 5.5% additional IOP reduction at month 1 and 8.8% additional IOP reduction at month 12. The IOP lowering efficacy of LBN was significant and persisted for 12 months. Visual field mean deviation, pattern standard deviation and retinal fiber layer thickness were not significantly decreased after switched to LBN 0.024% for 12 months, which is compatible with the results of the Jupiter study. Mean IOP reduction with LBN 0.024% in Jupiter study [8] was 4.3mmHg or 22% at 4 weeks and 5.3mmHg or greater than 25% at 52 weeks. 75% patients were NTG patients. It corresponded with the degree of IOP reduction to minimize glaucoma progression.

Subgroup analysis of POAG patients, it delivered a 32% mean IOP reduction from baseline with PGA use and 8.7% additional IOP reduction (total up to 10 mmHg mean IOP reduction) after switched to LBN at 12 months in our study. In APOLLO [5], LUNA [6] R and pooled analysis study [7], LBN delivered a 32% mean IOP reduction and up to 9 mmHg reduction from baseline in patients with open-angle glaucoma (OAG) and ocular hypertension (OHT) at month 3 and 12. Different population (European and African ancestry in APOLLO study) and glaucoma types may be the possible causes of different IOP lowering efficacy. Our study do not include ocular hypertension patients. In a retrospective chart review study [10], mean IOP lowering of 31% in treatment-naïve OAG patients with LBN at 4 months. Percent IOP lowering was 41% and 22% in POAG and NTG patients which was similar with results in the present

study. 32% and 19.6% mean IOP reduction from baseline with PGA and 9.3% and 5.4% mean additional reduction in POAG and NTG patients after switched to LBN for 3 months.

A network meta-analysis [9] for short-term efficacy of LBN for the treatment of OAG and OHT found that LBN significantly lowers IOP after 3 months' use compared to other treatments. LBN outperformed tafluprost (0.41 mm Hg) and travoprost (0.58 mm Hg) in mean difference IOP at 3 months. I n the present study, mean additional IOP reduction after LBN use was 0.6 mmHg in travoprost group and 1.4 mm Hg in tafluprost group at month 3. Mean IOP was significantly lowered after LBN use, demonstrating that LBN statistically outperformed tafluprost and travoprost throughout 12 months. The IOP reduction difference in tafluoprost group may due to different study design or population.

Approximately 40% of patients cannot tolerate the ocular adverse effects and dropped out at 1 month. The most common intolerable adverse effects were moderate to severe conjunctival hyperemia and eye irritation in the current study. The dropped out rate was higher than reported in the prior study. In Weinreb et al pooled analysis study [7], approximately one third of patients had conjunctival hyperemia at baseline, mostly mild to moderate, before treatment initiation. 29% subjects were treatment-naive and medication washout was performed in the remaining glaucoma patients. About 10% patients have moderate or severe conjunctival hyperemia and eye irritation was 8% in patients using LBN by week 6. Only 1.4% subjects discontinued due to ocular adverse effects in the pooled analysis study. In the VOYAGER study [11], the most common ocular treatment-related adverse effects was instillation site pain reported by 12.0% of subjects in the LBN 0.024% group, but did not affect compliance. Differences in the ethnic populations and different study design may have contributed to the greater dropped out rate observed in the present study. In a real-world study [10], the dropped out rate was 12.3% with use of LBN in treatment-naïve patients with open-angle glaucoma for 4 months. The higher dropped out rate in the present study may be due to direct switch from PG analog (travoprost and tafluprost) to LBN. Washout period may be important to increase the adherence. Mean age of patients in the present study (51.3 years) were younger than in the pooled analysis study [7] and Jupiter study [8] (64.9 and 62.5 years). Younger patients may have higher possibility to intolerant the ocular adverse effects due to the conjunctival hyperemia and ocular irritation.

In a recent meta-analysis [12], the percentage of patients who had at least one adverse event ranged from 24% in VOYAGER study [11] for 1 month, 13.4% and 23.8% in APOLLO [5] and LUNAR study [6] for 3 months to 21.6% in pooled study and 58.5% in JUPITER study [8] for 12 months. The three most common ocular side effects were conjunctival hyperemia, eye irritation, and dry eye in the meta-analysis study. In JUPITER study [8], conjunctival hyperemia (17.7%), growth eye lashes (16%), and irritation (11.5%) were the three most common ocular side effects. Conjunctival hyperemia occurred 2.8% and 9% for 3 months in APOLLO [5] and LUNAR study [6], and 5.9% for 1 year in Pooled analysis study [7]. The incidence of conjunctival hyperemia was less than in Asian-population of JUPITER study [8] and our study (13.6%). Conjunctival hyperemia have been reported about 4.9%-11.6% [13, 14] in tafluprost and about 42% in travoprost [15].

The percentage of tolerable ocular side effects were about 40.8% in the present study, which was similar to treatment-relative ocular adverse events reported in the Jupiter study (47.7%) [8]. Ocular adverse effects of special interest, like change in iris pigmentation and eyelash growth were uncommon, but eyelid pigmentation (4.9%) was noted in the present study. Similar with the result of Inone K et al [16], which reported there was no significant difference between the five types of Prostaglandin analogs with regards to eyelid pigmentation (4–6%), which included tafluprost and travoprost. Growth of eyelashes were significantly more frequent with travoprost and tafluprost (46%), which were much higher than LBN in JUPITER study [8].

The present study has several limitations. First, the sample of 81eyes was relatively small, and it may lead to a lack of statistical power. Subgroup analyses based on previous prostaglandin analog use or glaucoma type (POAG vs. NTG) may be limited by sample size, potentially resulting in insufficient power to detect meaningful differences between groups. Future prospective studies with larger samples and longer follow-up periods are warranted. Second, this study used retrospective design, suggesting that results cannot be generalized to other populations and that we cannot exclude the possibility of selection bias. Third, the high dropped out rate due to adverse effects warrant further investigations into patient-reported outcomes and quality of life measures.

## Conclusions

Although high intolerable adverse effects including conjunctival hyperemia and eye irritation happened in the first month, remaining sixty percent of patients exhibited statistically significant additional IOP reductions in the replacement of other PGAs during 12 months of clinical use of LBN 0.024% and also got benefits in the visual field and retinal nerve fiber layer thickness preservation.

## Supporting information

**S1 Checklist. Human participants research checklist.**
(DOCX)

**S1 Table. Mean additional intraocular pressure (IOP) reduction after switched to LBN in patients with primary open angle glaucoma (POAG) or normal tension glaucoma (NTG).**
(DOCX)

**S2 Table. Mean additional intraocular pressure (IOP) reduction in patients treated with tafluprost or travoprost before switched to LBN.**
(DOCX)

**S1 Data.**
(XLSX)

## Author Contributions

**Conceptualization:** Jou-Chen Huang.

**Data curation:** Chun-Mei Hsueh.

**Investigation:** Chen-Hsin Tsai.

**Methodology:** Chun-Mei Hsueh, Chen-Hsin Tsai.

**Project administration:** Si-Huei Lee.

**Software:** Jou-Chen Huang.

**Writing – review & editing:** Si-Huei Lee, Tsung-Jen Wang, Siao-Pei Guo.

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
