## [Decision Letter · Decision Letter 0]

25 Apr 2024

PONE-D-24-03946One-Year Experience with Latanoprostene Bunod Ophthalmic Solution 0.024% in Clinical Practice: A Retrospective Observational StudyPLOS ONE

Dear Dr. GUO,

Thank you for submitting your manuscript to PLOS ONE. After careful consideration, we feel that it has merit but does not fully meet PLOS ONE’s publication criteria as it currently stands. Therefore, we invite you to submit a revised version of the manuscript that addresses the points raised during the review process.

**The authors have drafted a thoughtful manuscript, however, it will benefit from the suggestions listed below.**

We look forward to receiving your revised manuscript.

Kind regards,

Natasha Gautam, MBBS, MS

Academic Editor

PLOS ONE

Journal Requirements:

Additional Editor Comments:

Please elaborate more on the adverse effects in results as well as discussion. I would suggest to group them into tolerable and intolerable side effescts (which lead to drop out at 1 month). Similarly in discussion, rather than phrasing the adverse effects as 2.5-19.8%, the authors should give the percentage of eyes which developed side effects, and also subgroup into tolerable and intolerable side effects as mentioned in results. They should also try to compare the side effects reported in literature for both latanoprostene bunod, as well as prostaglandin analogues used alone.

Reviewers' comments:

Reviewer's Responses to Questions

**Comments to the Author**

1. Is the manuscript technically sound, and do the data support the conclusions?

Reviewer #1: Partly

Reviewer #2: Partly

2. Has the statistical analysis been performed appropriately and rigorously? 

Reviewer #1: Yes

Reviewer #2: No

3. Have the authors made all data underlying the findings in their manuscript fully available?

Reviewer #1: Yes

Reviewer #2: Yes

4. Is the manuscript presented in an intelligible fashion and written in standard English?

Reviewer #1: Yes

Reviewer #2: No

5. Review Comments to the Author

**Reviewer #1**: In the methodology section kindly mention which measures were taken to ensure that the IOP readings on various visits were not affected due to diurnal variations. Also clarify that which reading was noted (tonoref or icare or average of both). In the result section, subgroup analyses based on previous prostaglandin analog use or glaucoma type (POAG vs. NTG) may be limited by sample size, potentially resulting in insufficient power to detect meaningful differences between groups. it's noteworthy that LBN was discontinued in a significant number of patients due to adverse effects after only one month of use, indicating a substantial rate of intolerance. The high rate of discontinuation due to adverse effects warrant further investigation and consideration. It would be valuable to compare the incidence and severity of adverse effects reported in this study with those reported in other similar studies to provide context and highlight any notable differences. Discuss how variations in patient demographics, disease characteristics, and clinical practices across different healthcare settings may impact the applicability of the results to broader patient populations. Offer suggestions for future research endeavors to address the study's limitations and expand upon its findings. This could include recommendations for prospective studies with longer follow-up periods, comparative effectiveness trials against other glaucoma therapies, or investigations into patient-reported outcomes and quality of life measures.

**Reviewer #2:** The presented study is an original retrospective observational research. Unfortunately, this article does not totally adhere to appropriate reporting guidelines and to the Quality Assessment Tool for Observational Research in PLOS ONE.

1. According to the style and format:

• Manuscript should be written in a clear, correct, and unambiguous language. Check it with proof reading of English grammar by using special “scientific editing service” or “manuscript editing service.” Please rewrite your manuscript, because some sentences are totally wrong. For example, in the “Abstract” Section the sentence “No significant changes in visual field and retinal nerve fiber layer thickness after 12 months follow up.” is an incomplete sentence that could not be accepted.

• Please include page numbers and continuous line numbers in the manuscript file and place each table in your manuscript file directly after the paragraph in which it is first cited.

2. The authors have not proposed a clear research question; the purpose should be corrected. It is not possible to evaluate the clinical use; the authors could evaluate the clinical efficacy or effectiveness. Please clarify the purpose: “To evaluate the efficacy of the first topical nitric oxide-donating prostaglandin analog (PGA), latanoprostene bunod (LBN) ophthalmic solution 0.024% (Vyzulta®), in intraocular pressure (IOP) reduction.”

3. In the “Methods” Section of the Abstract there is a contradiction: the least follow up mentioned to be 12 months and then it is written that “all included patients received LBN as a replacement for a PGA, with 1-, 3- , 6-and 12- month follow-up.” Please correct it and add a word “with examinations at 1-, 3-, 6- and 12-months follow-up.”

4. In the “Statistical analysis” Section and in the “Results” Section of Abstract and the manuscript please specify, which IOP values you use: mean or median? Furthermore, baseline IOP? IOP before LBN? PGA-IOP? Please clarify all these values.

5. Please correct “Methods” Section by adding the number of included patients (eyes), and then how many were excluded due to short follow up, and how many were left for analysis.

6. What does mean “PGA before LBN” in the Table 3 and Table 4?

7. There is a contradiction in the number of the patients with discontinued LBN at 1 month follow-up in the “Results” and “Discussion” Sections: 31 vs. 25 patients ….

6. PLOS authors have the option to publish the peer review history of their article (what does this mean?). If published, this will include your full peer review and any attached files.

Reviewer #1: No

Reviewer #2: No

---

## [Author Response · Author response to Decision Letter 0]

1 Jun 2024

Thank the editor and reviewers for the hard working to correct the mistakes and give us many useful suggestions for the article. Learn a lot and appreciate for all the help. Thank you so much!

---

## [Editor Report · Decision Letter 1]

6 Jun 2024

PONE-D-24-03946R1One-Year Experience with Latanoprostene Bunod Ophthalmic Solution 0.024% in Clinical Practice: A Retrospective Observational StudyPLOS ONE

Dear Dr. GUO,

Thank you for submitting your manuscript to PLOS ONE. After careful consideration, we feel that it has merit but does not fully meet PLOS ONE’s publication criteria as it currently stands. Therefore, we invite you to submit a revised version of the manuscript that addresses the points raised during the review process.

**The authors have addressed most of the comments nicely, however there are few suggestions as listed below.**

We look forward to receiving your revised manuscript.

Kind regards,

Natasha Gautam, MBBS, MS

Academic Editor

PLOS ONE

Journal Requirements:

Additional Editor Comments:

**The authors mentioned: Intraocular pressure measurements used non-contact tonometer (TONOREF III, NIDEK, Japan) and rechecked with iCare tonometer (TA01i) for 3 times. If two measurements are different, IOP data from iCare tonometer was recorded**.

**When IOP is being recorded using 2 different instruments, it is very rare that they would provide same IOP values. They are usually different by 1 or 2 points. Moreover 3 readings taken by icare alone would be different from each other. So it is still not clear, what equipments did the authors rely on everytime while checking IOP. From the above line, it appears that they used the NCT meausrement but in cases of discrepancy they relied on icare? Did they take the average of 3 readings from icare? If they relied on NCT during one visit, but there was discordance in IOP measurement between 2 equipments on subsequent visits, did they use icare that time? Does that mean that IOP values were not consistent and uniformly recorded over the visits? please elaborate**.

**A significant proportion of patients suffered from intolerable and tolerable side effects which lead to significant drop out, but it is not mentioned in the abstract results and conclusion. If there is a concern for word limit, the authors are advised to remove the lines "also benefits in the visual field and retinal nerve fiber layer thickness preservation" and instead focus on side effects to give the true picture to the readers.**

---

## [Author Response · Author response to Decision Letter 1]

12 Jun 2024

Dear editor Natasha Gautam: 

 Thank you for correcting the mistakes and reminding the important issue about the IOP measurements and the high intolerable adverse effects in the current study. Intraocular pressure measurements used non-contact tonometer (TONOREF III, NIDEK, Japan) and rechecked with iCare tonometer (TA01i) for 3 times. If IOP difference greater than 3 in three times measurements with non-contact tonometer, then recheck with iCare tonometer for three times. If the average values of two kinds of measurements are different, the average values of IOP data from iCare tonometer was recorded. The method of IOP measurement used for each patients was consistent across visits. 

 Sincerely yours, 

 Chunmei Hsueh and Siaopei Guo

---

## [Editor Report · Decision Letter 2]

1 Jul 2024

One-Year Experience with Latanoprostene Bunod Ophthalmic Solution 0.024% in Clinical Practice: A Retrospective Observational Study

PONE-D-24-03946R2

Dear Dr. GUO,

We’re pleased to inform you that your manuscript has been judged scientifically suitable for publication and will be formally accepted for publication once it meets all outstanding technical requirements.

Kind regards,

Natasha Gautam, MBBS, MS

Academic Editor

PLOS ONE
---

## [Editor Report · Acceptance letter]

8 Jul 2024

PONE-D-24-03946R2 

PLOS ONE

Dear Dr. GUO, 

I'm pleased to inform you that your manuscript has been deemed suitable for publication in PLOS ONE. Congratulations! Your manuscript is now being handed over to our production team.

Kind regards, 

on behalf of

Dr. Natasha Gautam 

Academic Editor

PLOS ONE